Associations between physical activity and emotional and behavioural problems in Chinese children and adolescents with intellectual disabilities

Zhong Yecheng
Zhou Junjie
Li Niuniu
Xu Wenhong
Qi Jing qijing@zjnu.edu.cn
College of Physical Education and Health Sciences, Zhejiang Normal University , Jinhua, Zhejiang , China
Khoo Selina
Electronic publication date: 2025 Feb 14
Publication date: 2025
Volume: 13
Electronic Location ID: e18949
Received 2024 Aug 29; Accepted 2025 Jan 16
Copyright: © 2025 Zhong et al.
Copyright year: 2025
Copyright holder: Zhong et al.
License: This is an open access article distributed under the terms of the Creative Commons Attribution License, which permits unrestricted use, distribution, reproduction and adaptation in any medium and for any purpose provided that it is properly attributed. For attribution, the original author(s), title, publication source (PeerJ) and either DOI or URL of the article must be cited.
License URL: https://creativecommons.org/licenses/by/4.0/

Keywords: Intellectual disability, Children and adolescents, Moderate-to-vigorous physical activity, Emotional problems, Behavioural problems

Funding: Education Sciences Planning of Zhejiang Province, China 2023SCG375 This work was supported by the Education Sciences Planning of Zhejiang Province, China (No. 2023SCG375), under the project titled “The assessment and physical activity intervention of children and adolescents with disabilities based on their 24-hour movement”. The funders had no role in study design, data collection and analysis, decision to publish, or preparation of the manuscript.

==============================
Background and purpose

Emotional and behavioural problems (EBPs) are the two main forms of mental health problems that hinder the social interactions, learning and daily living of children and adolescents with intellectual disabilities (ID). Studies have shown that sufficient moderate-to-vigorous physical activity (MVPA) is associated with mental health outcomes among children and adolescents with typically developing (TD). However, these associations have not been fully studied in children and adolescent with ID, especially in samples from China. Therefore, this study aimed to examine the associations between MVPA and EBPs in Chinese children and adolescents with ID.

Methods

A cross-sectional survey was conducted with 116 students with ID (35.34% girls) aged 6–18 years. The wGT3-BT accelerometer was used to measure physical activity (PA), and the Chinese version of the parent-rated Strengths and Difficulties Questionnaire (SDQ) was used to assess the participants’ EBPs. A series of binary logical regression analyses was conducted to examine the ability of the MVPA guideline attainment in predicting EBPs in the total sample.

Results

The descriptive results showed that 43.97% of children and adolescents with ID present with EBPs. After controlling for age, sex, ID severity and weight status, logistic regression analysis indicated that the participants who meet the MVPA guideline exhibited significantly lower odds ratio for emotional symptoms (OR = 0.334, 95% CI [0.114–0.975], p = 0.045), peer problems (OR = 0.071, 95% CI [0.015–0.328], p < 0.001) and total difficulties (OR = 0.192, 95% CI [0.069–0.535], p = 0.002) compared with those who did not meet the guidelines.

Conclusion

The prevalence of EBPs in Chinese children and adolescents with ID remains high. Children and adolescents with ID meeting the MVPA guideline were more likely to have lower risks for emotional symptoms, peer problems and total difficulties than those who did not meet the guidelines. Future research adopting longitudinal or interventional designs to determine the relationships between PA and EBPs in this population are expected.

Introduction

Intellectual disability (ID) refers to a condition characterised by significant limitations in cognitive functioning and adaptive behaviour, and symptoms usually emerge before the age of 22 years (American Association on Intellectual and Developmental Disabilities, 2021). When compared with typically developing (TD) peers, children and adolescents with ID exhibit about three to four times higher rate of mental health problems (Allerton, Welch & Emerson, 2011; Yang, Liang & Sit, 2022). In China specifically, data from 12 provincial administrative regions showed that 39.39% of children and adolescents with ID exhibited problem behaviours during the COVID-19 pandemic, such as hyperactivity, stereotyped behaviour and inappropriate language (Ma et al., 2022). The high prevalence of these problems may be due to individuals with ID generally having more difficulties in generalising information, learning from past experiences and misinterpreting what is expected of them (Krebs, 2005).

Emotional and behavioural problems (EBPs) is referred to the group of disorders classified according to the 10th revision of International Classification of Diseases as “Behavioural and emotional disorders with onset usually occurring in childhood and adolescence” (F90-F98) including hyperkinetic disorders, conduct disorders, emotional disorders and disorders of social functioning (World Health Organization, 2019; Kosticova, Husarova & Dankulincova, 2020). EBPs interfere with social interactions, parent-child relationships, learning and daily living in individuals with ID, and often persist into adulthood (Gleason et al., 2016; Lee & Jeoung, 2016). In addition, children’s EBPs also result in family breakdown, divorce and long-term unemployment as a result of constantly leaving children at home (Rey, 2015; Alenko et al., 2020). Studies have shown that parents of children with ID suffer from high levels of parenting pressure (Hastings & Beck, 2004; Halvorsen et al., 2019), which also brings instability to the harmonious development of society. Overwhelming evidence shows that physical activity (PA) is associated with mental health outcomes in children with TD (Bell et al., 2019; Booth et al., 2023; Asare et al., 2023), and this relationships may be more consistent and robust for higher-intensity PA, such as moderate-to-vigorous PA (MVPA) (Poitras et al., 2016). However, such evidence in TD adolescents is limited by methodological weaknesses, most notably over-reliance on self-reported measures of MVPA (Sagatun et al., 2007; Kantomaa et al., 2008; Tandon et al., 2021). Despite self-reported measures being an ideal method for large-scale monitoring and surveillance, owing to their lower testing burden and costs, they are thought to have some issues, such as recall bias and high error rates (Leung, Siebert & Yun, 2017). Thus, accelerometer-measured PA levels are required to improve the research quality in this field. In children and youth with ID, two studies investigated the associations between PA and EBPs, but the results were inconsistent. For example, Fiorilli et al. (2016) compared EBPs between ID participants in the Special Olympics and those who had never participated in the Special Olympics, and their results indicated that participation in sports activities was associated with improved well-being, behaviours and social skills. Meanwhile, Whitney et al. (2019) found that PA was not related to depression or anxiety problems in children and adolescents with ID.

According to the World Health Organization (2010), children and youth with and without disabilities should engage in at least 60 min/day of MVPA to gain health benefits. However, the presence of disability and associated conditions can limit one’s PA participation (Rimmer et al., 2004). A previous meta-analysis showed that youth with disabilities were less active than their healthy counterparts due to disability-related limitations, including those with ID (Jung et al., 2018). Evidence also shows that PA levels decline with age in individuals with ID (Shields, Dodd & Abblitt, 2009; Yuan et al., 2022a), and this trend continues from adolescence into adulthood (Borland et al., 2020). Considering the numerous psychological benefits of PA participation and the high prevalence of EBPs in children and adolescents with ID, more consistent evidence is needed to clarify the relationships between PA levels and EBPs in this population.

China is the world’s most populous country with a population of more than 1.41 billion. On the basis of the national population, the prevalence of ID in China was 7.5 per 1,000 children (Wu et al., 2010). The high prevalence of EBPs in this population has garnered the wide attention of Chinese families, schools and society (Kwok, Cui & Li, 2011; Lin & Lin, 2011). An essential element in determining whether behaviours should be targeted for intervention is determining the associations between the behaviours and health outcomes (Sallis, Owen & Fotheringham, 2000). However, the relationships between PA and EBPs in previous research have not been fully investigated in Chinese children and youth with ID. Thus, studying these relationships can provide public health insights for improvement of EBPs among individuals with ID in China.

Using accelerometer-measured data, the aims of this study were to (1) investigate the MVPA level and the prevalence of EBPs of Chinese children and adolescents with ID, as well as the demographic factors related to their MVPA and EBPs, and (2) examine the associations of MVPA with EBPs in this target population, while taking age, sex, ID severity and weight status into consideration.

Materials and Methods

Participants

A total of 168 children and adolescents were recruited from a special school in a city in northwestern China. The inclusion criteria for all participants were as follows: (1) with a physician diagnosis of ID in accordance with the fifth edition of the Diagnostic and Statistical Manual of Mental Disorders (DSM-5) (American Psychiatric Association, 2013); (2) aged 6–18 years (Xu et al., 2018; Yuan et al., 2022b); (3) able to follow instructions and communicate verbally; (4) without mobility issues that would limit activity; and (5) had valid and complete measurement data. Ineligible participants were excluded based on these screening questions.

Measures

Anthropometric and demographic profiles

Body height (cm) and weight (kg) were measured by well-trained research staff using a portable instrument (GMCS-IV; Jianmin, Beijing, China). Then, body mass index (BMI; kg/m2) was calculated. According to age- and sex-specific cut-offs of overweight and obesity for Chinese school-aged children and adolescents (National Institute for Nutrition and Health Chinese Center for Disease Control and Prevention, 2018), the participants were divided into overweight (≥85th and <95th percentile)/obese (≥95th percentile) and non-overweight/obese.

Demographic information (e.g., age, sex, ID severity) was obtained from the participants’ parents. Age was treated as a dichotomised variable with children (6–12 years) and adolescents (13–18 years) in the data analysis (Xu et al., 2018; Shen et al., 2021).

Physical activity

The wGT3-BT accelerometer (ActiGraph, Pensacola, FL, USA) was used to assess physical activity, using its official software (ActiLife) for data initialisation and download. This instrument has been widely used in children and adolescents with ID (McGarty, Penpraze & Melville, 2016; Xu & Wang, 2023). The accelerometers were initialised with a sampling frequency of 30 Hz. The participants were instructed to wear an accelerometer at the waist on an elasticised belt during waking hours (except swimming and showering) for seven consecutive days. Continuous counts of zero for ≥20 min were defined as non-wear time (Choi et al., 2011). A total of 600 min (10 h) of waking wear time per day or more were considered valid (Anderson, Hagströmer & Yngve, 2005). At least three valid weekdays and one valid weekend day of objective data were needed for the final analysis. MVPA was output in counts/15 s and defined as all activity ≥2,296 counts/min (Evenson et al., 2008). Evenson’s cutting points in this work were used and validated on children and youth with disabilities (Downs et al., 2016; Sit et al., 2017).

Emotional and behavioural problems

The Chinese version of the parent-rated Strengths and Difficulties Questionnaire (SDQ) (Goodman, 1997) was used to assess the participants’ EBPs, which showed good validity and reliability (Cronbach’s alpha = 0.81; intra-class correlation = 0.80; the discriminative validity = 0.84) (Lai et al., 2010). The SDQ records positive and negative behavioural attributes and generates scores for clinically relevant aspects. SDQ has been used to assess behavioural problems in children and youth with ID (Murray, Hastings & Totsika, 2021). The questionnaire consisted of 25 items divided into five subscales, each with five items: emotional symptoms, conduct problems, hyperactivity/inattention, peer problems and prosocial behaviour. The first four subscales of the 20 items asked about negative behavioural attributes (‘difficulties’) and the final scale (five items) asked positive attributes (‘strengths’). Each item is scored on a 0–2 scale (not true, somewhat true, certainly true). The sum of the first four subscales generates a total difficulty score ranging from 0–40. Higher scores reflect greater behaviour problems. The other five positively worded questions of the prosocial behaviour scale are reverse scored: a higher score on this scale indicates a higher likelihood of engaging in voluntary and prosocial actions. Each respondent’s total difficulties and prosocial scores were classified as ‘abnormal’ and ‘normal’ (including borderline group) in accordance with the cutoff thresholds validated for Chinese children (Du, Kou & Coghill, 2008). For the raw score of the total difficulties, 0–14, 15–16 and 17–40 were rated as normal, borderline and abnormal, respectively. Meanwhile, raw scores of 10–6, 5 and 4–0 in the scale of prosocial behaviour were rated as normal, borderline and abnormal, respectively.

Procedures

Ethical approval (ZSRT2024106) was obtained from the Ethics Committee of Zhejiang Normal University. Prior to the data collection, written consent forms were distributed to the students’ parents or their guardians. The first author and two graduate students conducted the survey and test. The participants were instructed to wear the accelerometers for seven consecutive days and follow their normal daily routines. Written instructions reminding the students and their parents or guardians to wear the accelerometers were also provided to class teachers to increase compliance. The researchers checked how well the ActiGraph devices were worn each morning and took them back a week later. The parents or guardians were asked to complete the questionnaires on children’s emotional and behavioural problems when they came to the school to pick up their children.

Statistical analyses

Data were analysed using SPSS 27.0 (IBM Corp., Armonk, NY, USA). Descriptive statistics, including means, standard deviations, frequencies, and percentages, were obtained for variables where appropriate. The chi-square test was used to compare group differences (e.g., boy vs. girls; moderate ID vs. severe ID; non-overweight/obese vs. overweight/obese) in the sample size distribution and the number of students who met or did not meet the MVPA guideline. Group differences in height, weight, and BMI were compared by sex using the independent t test. A series of binary logical regression analyses was conducted to examine the ability of the MVPA guideline attainment in predicting EBPs after controlling for age, sex, ID severity and weight status. Statistical significance was set at p < 0.05 for all tests.

Results

Characteristics of the participants

Of the 168 students initially recruited in this study, 52 were excluded from data analyses (five students were over 18 years old; eight had mobility issues; 39 did not have valid accelerometric data due to their poor compliance with the wearing protocol), thus resulting in a final dataset of 116 students.

Table 1 shows the characteristics of the participants included in the data analyses. The sample consisted of 64.66% boys and 35.34% girls, with a mean age of 13.11 ± 3.05 years. The mean BMI percentile was 20.32 ± 4.29, and no significant differences were found for BMI by sex (p = 0.915). The percentage of the participants with moderate and severe ID was 31.03% and 68.97%. In all the samples, more than 34% of the participants were classified as overweight/obese, but no significant difference in weight status was found between boys and girls (p = 0.642).

Table 1 Characteristics of the participants.

Variables	Mean (SD)	Sex difference
p-value	
Total (n = 116)	Boys (n = 75, 64.66%)	Girls (n = 41, 35.34%)	
Age (year)	13.11 (3.05)	13.05 (3.22)	13.22 (2.76)	0.781	
Children (6–12 y), n (%)	54 (46.55)	37 (49.33)	17 (41.46)		
Adolescents (13–18 y), n (%)	62 (53.45)	38 (50.67)	24 (58.54)	0.417	
Height (cm)	151.04 (16.69)	152.99 (17.85)	147.49 (13.82)	0.090	
Weight (kg)	47.67 (16.65)	49.00 (17.33)	45.22 (15.21)	0.244	
BMI (kg/m2)	20.32 (4.29)	20.29 (3.84)	20.38 (5.06)	0.915	
ID severity, n (%)					
Moderate	36 (31.03)	26 (34.67)	10 (24.39)		
Severe	80 (68.97)	49 (65.33)	31 (75.61)	0.253	
Weight status, n (%)					
Non-overweight/obese	76 (65.52)	48 (64.00)	28 (68.29)		
Overweight/obese	40 (34.48)	27 (36.00)	13 (31.71)	0.642	
Note:

BMI, body mass index; ID, intellectual disability.

Group differences in MVPA

Table 2 presents the descriptive statistics and group comparison of the participants’ MVPA. Overall, children and adolescents with ID spent 48.09 ± 32.05 min of MVPA per day. Only 46 participants (39.66%) met the World Health Organization’s (WHO) PA recommendation. No significant group difference for age, sex and weight status was found between participants who met the PA guideline and those who did not meet the guideline. Meanwhile, the participants with moderate ID had a significantly higher compliance rate of PA guideline than those with severe ID (χ2 = 15.916, p < 0.001).

Table 2 Descriptive statistics and group comparison in MVPA.

Demographic factors	MVPA time, n (%)	χ2 value	p-value	
<60 min/d	≥60 min/d	
Total	70 (60.34)	46 (39.66)			
Age					
Children (6–12 y)	35 (64.81)	19 (35.19)	0.844	0.358	
Adolescents (13–18 y)	35 (56.45)	27 (43.55)			
Sex					
Boys	46 (61.33)	29 (38.67)	0.087	0.768	
Girls	24 (58.54)	17 (41.46)			
ID severity					
Moderate	12 (33.33)	24 (66.67)	15.916	<0.001	
Severe	58 (72.50)	22 (27.50)			
Weight status					
Non-overweight/obese	49 (64.47)	27 (35.53)	1.570	0.210	
Overweight/obese	21 (52.50)	19 (47.50)			
Note:

MVPA, moderate-to-vigorous physical activity; ID, intellectual disability.

Distribution of emotional and behavioural problems

The prevalence of EBPs, stratified by age, sex, ID severity and weight status, is reported in Fig. 1. In total, 43.97% of participants with ID present with EBPs. For subfields of EBPs, abnormal prosocial behaviour and peer problems were the most common problems, accounting for 41.38% and 37.07%. More ID adolescents reported emotional symptoms (χ2 = 4.796, p = 0.029) and prosocial behaviour (χ2 = 4.568, p = 0.033), than did ID children. The participants with moderate ID had more difficulties with hyperactivity/inattention (χ2 = 9.985, p = 0.002) and peer problems (χ2 = 10.118, p = 0.001), and less prosocial behaviour (χ2 = 4.325, p = 0.038) than those with severe ID. In addition, individuals who were overweight/obese showed more conduct problems (χ2 = 5.513, p = 0.019) than those who were non-overweight/obese.

Figure 1 Prevalence of emotional and behavioural problems stratified by sex (A), age (B), ID severity (C) and weight status (D) in children and adolescents with ID.

* p < 0.05; ** p < 0.01. Raw score was rated as abnormal: total difficulties score, 17–40; emotional symptoms, 5–10; conduct problems, 4–10; hyperactivity/inattention, 8–10; peer problems, 6–10; prosocial behaviour, 4–0.

Associations of MVPA with emotional and behavioural problems

Figure 2 presents the associations of PA guideline attainment with EBPs in the overall sample. After controlling for age, sex, ID severity and weight status, the participants who meet the PA guideline exhibited significantly lower odds ratio for emotional symptoms (OR = 0.334, 95% CI [0.114–0.975], p = 0.045), peer problems (OR = 0.071, 95% CI [0.015–0.328], p < 0.001) and total difficulties (OR = 0.192, 95% CI [0.069–0.535], p = 0.002) compared with those who did not meet the guidelines.

Figure 2 Odds ratios for PA guideline attainment associated with EBPs in children and adolescents with ID.

Models adjusted for age, sex, ID severity and weight status. MVPA, moderate-to-vigorous physical activity; OR, odds ratio; CI, confidence interval.

Discussion

The first purpose of this study was to investigate the MVPA level and the prevalence of EBPs in Chinese children and adolescents with ID, as well as the demographic factors related to their MVPA and EBPs. Our results showed that 39.66% of Chinese children and adolescents with ID met the WHO’s PA guideline, is similar to the PA guideline attainment reported in other published studies (Shields, Dodd & Abblitt, 2009; Izquierdo-Gomez et al., 2014). Among the demographic factors, we found that the participants with moderate ID are more likely to meet the PA guideline, compared with those with profound ID. Yet, a study by Yuan et al. (2022a) found no significant differences in PA levels in the participants with different levels of ID. This inconsistency may lie in the fact that different measures of PA were used. The previous study used questionnaires to assess participants’ PA levels, while the current study adopted accelerometers to measure them.

We also found that 43.97% of Chinese children and adolescents with ID present with EBPs. The prevalence found in this study is similar with level of prevalence reported in the previous review (38–49%) (Buckley et al., 2020). Notably, the prevalence of EBPs in the current study was even higher than that among this population during the COVID-19 pandemic in China (39.39%) (Ma et al., 2022). Indeed, China has paid great attention to the mental health of children and adolescents at the national level (The State Council of the People’s Republic of China, 2019). Mental health education in primary and secondary schools has been vigorously promoted. However, due to the influence of some factors, such as faculty structure, management evaluation and home-school cooperation, there are still some deficiencies in mental health education in special schools (Gao et al., 2013). In our study, more adolescents with ID reported emotional symptoms and prosocial behaviour than did children with ID. This finding is understandable. The transition from childhood to adolescence is a stage when young people are most vulnerable to developing mental health problems. Individuals with ID may not be fully aware of the process of transition, but still often worry about their physical appearance and are anxious about their future (Graham, 1991; Gobrial & Raghavan, 2012). As children with ID grow, they may be better able to express their emotions, their parents become better observers of emotional symptoms or these symptoms become more prominent at home (Weisbrot et al., 2005). This may be another reason why adolescents with ID have more prominent emotional problems than children do. Consistent with previous findings (Wagemaker, Hofmann & Müller, 2023), the increased prosocial behaviour in adolescents with ID compared to children may be due to their evolving social cognition with the process of growth (Smogorzewska, Szumski & Grygiel, 2019). Further, ID adolescents are more eager to pursue prosocial goals and contribute to society than children with ID, which motivates them to participate more actively in social life and develop more prosocial behaviours (Fuligni, 2019).

Compared with the participants with severe ID, those with moderate ID were found to have more hyperactivity/inattention and peer problems, and less prosocial behaviour. Individuals with severe ID often lack social, learning and daily living skills, they tend to be supervised by their parents or guardians for a great protection (Chadwick et al., 2005). This may cause them to exhibit fewer hyperactivity/inattention and peer problems than those with moderate ID. Interestingly, findings from the current study conflict somewhat with recent research conducted by Wagemaker, Hofmann & Müller (2023), in which students with ID with greater general functioning display more prosocial behaviour. This discrepancy may be attributed to the difference in measurements. Wagemaker, Hofmann & Müller’s (2023) research used teacher-rated SDQ to assess the participants’ EBPs, while the current study used parent-reported questionnaires. More research is necessary to examine the effect of ID severity on prosocial behaviour.

In terms of weight status, our study showed that ID participants who were overweight/obese present more conduct problems than those who were non-overweight/obese. This finding is in line with that in children and adolescents with TD (Slykerman et al., 2020; Beynon, 2023). Children with obesity usually experience more discrimination, bullying and even social isolation than their peers with normal weight, which may have a negative impact on their mental health (Sagar & Gupta, 2018).

A secondary purpose of this study was to examine the associations of MVPA with EBPs in children and adolescents with ID. After controlling for all the covariates (i.e., age, sex, ID severity and weight status), we found that MVPA may be an effective predictor of EBPs among ID participants. Specifically, participants who met the PA guideline had significantly lower odd ratios for emotional symptoms, peer problems and total difficulties compared to those who did not meet the guideline. A similar correlation between PA and emotional symptoms was also found in a previous study on children with attention-deficit/hyperactivity disorder, in which reduced symptoms of anxiety or depression were identified if these children participated in PA three or more times per week (Kiluk, Weden & Culotta, 2009). MVPA is a strong predictor of emotional symptoms and can be explained by neurobiological mechanisms (Matta Mello Portugal et al., 2013; Amatriain-Fernández et al., 2021). Firstly, regular participation in PA can increase individuals’ levels of endocannabinoids, thereby reducing sensitivity to pain (Amatriain-Fernández et al., 2021). Secondly, PA training helps stimulate the synthesis and release of the major central nervous system neurotransmitters, such as serotonin, dopamine and norepinephrine, which are beneficial for regulating one’s anxiety level, alertness state, and the pleasure and reward system (Matta Mello Portugal et al., 2013). Finally, the brain-derived neurotrophic factor (BDNF) is associated with neurogenesis and angiogenesis (Deslandes et al., 2009). The PA participation is thought to contribute to the formation and release of BDNF, which acts like a regular antidepressant and may have a positive effect on emotional symptoms (Russo-Neustadt, 1999; Deslandes et al., 2009). Interestingly, our findings conflict somewhat with the previous study (Whitney et al., 2019), in which they found that PA was not related to the ID children’s anxiety or depression problems. Possible reasons for this inconsistency may be due to different measures of PA and EBPs. The previous study measured PA by parent-reported questionnaires and targeted a specific mental health problem such as depression, whereas the current study used accelerometers to estimate the PA levels and captured a multi-dimensional condition of EBPs.

In this study, those who met the PA recommendations had significantly fewer peer problems than those who did not meet the recommendations. Similar findings were found in children and youth with TD. For example, a longitudinal study on Dutch children found that higher MVPA levels at age 5–6 were associated with fewer peer problems at age 10–11 (Yang, Corpeleijn & Hartman, 2023). A study on 6,135 children in the UK found that an increase in PA at age 7 was associated with a decrease in peer problems at age 11 (Ahn et al., 2018). Regular PA participation offers opportunities for children and youth to develop social skills and improve social competence, thereby enhancing peer relationships and reducing the risk of social isolation (Eime et al., 2013; Werneck et al., 2019). In addition, schools are the main environment for children and youth with disabilities to accumulate MVPA (Einarsson et al., 2016). Being physically active in a school environment can provide ID students with familiar classmates or friends, allowing them to share experiences and goals, and continue to deepen friendships (Eccles et al., 2003).

Strengths and limitations

Using accelerometers to assess PA levels and parent-rated SDQ to investigate EBPs, this study expanded the evidence of associations between PA and EBPs in children and adolescents with ID. Our findings have implications for service planning, resource allocation, and prevention and treatment programming for students with ID in China. Some limitations, however, should also be mentioned. First, the participants were purposely sampled, limiting the generalisability of our findings. Second, the cross-sectional survey design of this study precludes conclusions regarding causality between PA level and EBPs in children and youth with ID. Finally, the utilization of parent-report questionnaires on the participants’ EBPs may be viewed as a limitation. More specifically, it is preferable to use several informants, such as parents and teachers, to measure mental health among children and adolescents (Sagatun et al., 2007).

Conclusions

Despite growing concern and awareness, the prevalence of EBPs in Chinese children and adolescents with ID remains high. ID students who meet the MVPA guideline were more likely to have lower risks for emotional symptoms, peer problems and total difficulties than those who did not meet the guidelines. Therefore, meeting the MVPA guideline should be considered an effective intervention and can inform the design of strategies for the prevention of mental health problems in children and youth with ID.

Supplemental Information

Supplemental Information 1 Raw data.

Supplemental Information 2 STROBE Checklist.

We would like to thank the students, parents, and staff for their participation, enthusiasm and support.

Additional Information and Declarations

Competing Interests

The authors declare that they have no competing interests.

Author Contributions

Yecheng Zhong performed the experiments, analyzed the data, prepared figures and/or tables, authored or reviewed drafts of the article, and approved the final draft.

Junjie Zhou performed the experiments, analyzed the data, prepared figures and/or tables, and approved the final draft.

Niuniu Li performed the experiments, prepared figures and/or tables, and approved the final draft.

Wenhong Xu conceived and designed the experiments, authored or reviewed drafts of the article, and approved the final draft.

Jing Qi conceived and designed the experiments, performed the experiments, authored or reviewed drafts of the article, and approved the final draft.

Human Ethics

The following information was supplied relating to ethical approvals (i.e., approving body and any reference numbers):

Recruitment and data collection procedures were approved by the Human Research Ethics Committee (No: ZSRT2024106) of Zhejiang Normal University.

Data Availability

The following information was supplied regarding data availability:

The raw data are available in the Supplemental File.

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
