# Peer review of "Associations between physical activity and emotional and behavioural problems in Chinese children and adolescents with intellectual disabilities"

_PeerJ, doi:10.7717/peerj.18949_

## Round 0.1 · original submission · Major Revisions

Thank you for your submission. The reviewers have identified a number of concerns that must be addressed especially the novelty of the research.

Reviewer 1 ·

Basic reporting

The article is well structured, with a current and relatively comprehensive bibliography for the proposed topic of the article.
Perhaps it would have been necessary to define Emotional and Behavioral Problems with more bibliographic references, perhaps even with reference to definitions adapted to the diagnostic criteria, for example from the DSM5 or ICD.
The research question should be formulated and well highlighted in order to see in the conclusions how the study responded to it.

Experimental design

The article has a well-defined research purpose and has a good structuring of the content. It presents the description of the research methodology and respects the principles of ethics of research with human subjects.

Validity of the findings

I believe that the conclusions should be somewhat more elaborate and refer to other similar research on this topic, briefly presenting similarities and differences between the results obtained. It would be worthwhile to make a reference to the attainment of the goal, to the hypotheses of the research.

·

Basic reporting

The manuscript is written in clear and professional English, adhering to standards of technical accuracy and courtesy. While the article provides sufficient background and references relevant literature, there is room for improvement in contextualizing its findings within the broader field. The structure follows a standard format with appropriate tables, though the absence of figures impacts the clarity of data presentation. All necessary raw data appears to be provided. Overall, the manuscript is self-contained and addresses the hypotheses adequately, without unnecessary subdivision of content.

Experimental design

The research question is well-defined and relevant, addressing a meaningful topic related to physical activity and mental health in children with intellectual disabilities (ID). However, the study's contribution to the knowledge gap is somewhat limited, as much of the work reiterates known findings from typically developing children (see details in the later section). The investigation is technically sound and appears to adhere to ethical standards. The methods are described with sufficient detail, allowing for potential replication of the study by other researchers.

Validity of the findings

The findings are based on robust, statistically sound data, with appropriate controls. While the study does not introduce highly novel insights, its focus on children with intellectual disabilities in China adds value. All underlying data are provided, and the conclusions are appropriately linked to the research question and supported by the results.

Additional comments

The relationship between physical activity and mental health has been widely studied in both children with and without disabilities. The novelty of this manuscript appears limited, as its main contribution is focusing on children with intellectual disabilities (ID) in China. As the authors themselves note, “Overwhelming evidence shows that physical activity (PA) is associated with mental health outcomes among children and adolescents with TD” (Hosker, Elkins & Potter, 2019; Biddle et al., 2019). Given that these studies explore similar themes, it would be essential for the authors to better contextualize its findings within the existing literature to highlight any unique contributions or insights.

There are no figures in the manuscript, which could make the results harder to interpret for readers. The tables are descriptive, but the inclusion of visual aids like graphs would enhance the clarity of findings. For example, a graph showing the differences in EBPs between those who meet the MVPA guideline and those who don't could make the results more compelling.

Table 1

1. Table 1 lacks p-values or test results to support the claim of "no significant differences." These should be included for transparency and to validate the findings.

2. In addition, potential confounding factors like socioeconomic status (SES) and comorbidities (e.g., autism, ADHD) could be considered. Controlling some additional confounding factors is recommended. However, it’s understandable given the potential difficulty of data accessibility.

Table 4
The model adjusts for essential factors (age, gender, ID severity, and weight status), which are likely confounders in this context. However, along with table 1 comments, the authors could also consider additional controls, such as socioeconomic status (SES) or comorbidities, which might influence both MVPA levels and emotional/behavioral problems.

---

## Round 0.2 · accepted · Accept

Thank you for your revised submission. I am satisfied that you have addressed the remaining concerns of the reviewers, and am happy to accept your paper for publication.

·

Basic reporting

no comment

Experimental design

no comment

Validity of the findings

no comment

Additional comments

The revised manuscript has addressed my initial concerns and questions from the first review. The authors have improved the contextualization of their findings within the existing literature, clarified statistical analyses by including p-values in Table 1, and enhanced data presentation by replacing tables with figures for better clarity. While some limitations remain due to data accessibility challenges, these are transparently acknowledged, and the overall quality of the manuscript has significantly improved. I recommend accepting the revised manuscript for publication.